# Computational phenotyping of brain-behavior dynamics underlying approach-avoidance conflict in major depressive disorder

Mads L. Pedersen[1,2,3]*, Maria Ironside[4,5], Ken-ichi Amemori[6,7,8,9], Callie L. McGrath[4,5], Min S. Kang[5], Ann M. Graybiel[6,7], Diego A. Pizzagalli[4,5,10‡], Michael J. Frank[1,2‡]*

1 Department of Cognitive, Linguistic & Psychological Sciences, Brown University, Providence, Rhode Island, United States of America, 2 Carney Institute for Brain Science, Brown University, Providence, Rhode Island, United States of America, 3 Department of Psychology, University of Oslo, Oslo, Norway, 4 Department of Psychiatry, Harvard Medical School, Boston, Massachusetts, United States of America, 5 Center for Depression, Anxiety and Stress Research, McLean Hospital, Boston, Massachusetts, United States of America, 6 Department of Brain and Cognitive Sciences, Massachusetts Institute of Technology, Massachusetts, United States of America, 7 McGovern Institute for Brain Research, Massachusetts Institute of Technology, Cambridge, Massachusetts, United States of America, 8 Hakubi Center for Advanced Research, Kyoto University, Kyoto, Japan, 9 Primate Research Institute, Kyoto University, Aichi, Japan, 10 McLean Imaging Center, McLean Hospital, Boston, Massachusetts, United States of America

‡ These authors are joint senior authors on this work.
* madslupe@gmail.com (MLP); michael_frank@brown.edu (MJF)

**Data Availability Statement:** Data are available at https://github.com/madslupe/AAC_DDM_MDD/.

## Abstract

Adaptive behavior requires balancing approach and avoidance based on the rewarding and aversive consequences of actions. Imbalances in this evaluation are thought to characterize mood disorders such as major depressive disorder (MDD). We present a novel application of the drift diffusion model (DDM) suited to quantify how offers of reward and aversiveness, and neural correlates thereof, are dynamically integrated to form decisions, and how such processes are altered in MDD. Hierarchical parameter estimation from the DDM demonstrated that the MDD group differed in three distinct reward-related parameters driving approach-based decision making. First, MDD was associated with reduced reward sensitivity, measured as the impact of offered reward on evidence accumulation. Notably, this effect was replicated in a follow-up study. Second, the MDD group showed lower starting point bias towards approaching offers. Third, this starting point was influenced in opposite directions by Pavlovian effects and by nucleus accumbens activity across the groups: greater accumbens activity was related to approach bias in controls but avoid bias in MDD. Cross-validation revealed that the combination of these computational biomarkers were diagnostic of patient status, with accumbens influences being particularly diagnostic. Finally, within the MDD group, reward sensitivity and nucleus accumbens parameters were differentially related to symptoms of perceived stress and depression. Collectively, these findings establish the promise of computational psychiatry approaches to dissecting approach-avoidance decision dynamics relevant for affective disorders.

**Funding:** The present work was partially supported by the Research Council of Norway 262372 to MLP, National Institute of Mental Health grants R01 MH084840 and R01MH115905- 01 to MJF, R01 MH108602, R37 MH068376, and R01 MH101521 to DAP, the Saks Kavanaugh Foundation, NIH/NINDS grant R01 NS025529 and Amy Research Office grant W911NF-16-1-0474 to AMG, a Kaplen Fellowship on Depression and Livingston Fellowship to CLM., and MEXT KAKENHI (18H05131, 18K19497) grants to KIA. MI and CLM were partially supported by the John and Charlene Madison Cassidy Fellowship in Translational Neuroscience. The funders had no role in study design, data collection and analysis, decision to publish, or preparation of the manuscript.

**Competing interests:** I have read the journal's policy and the authors of this manuscript have the following competing interests: MJF receives consulting fees from F-Hoffman LaRoche Pharmaceuticals for activities unrelated to this project. Over the past three years, DAP received funding from NIMH, Brain and Behavior Research Foundation, the Dana Foundation, and Millennium Pharmaceuticals; consulting fees from Akili Interactive Labs, BlackThorn Therapeutics, Compass Pathway, Boehringer Ingelheim, Outsuka Pharmaceuticals, and Takeda Pharmaceuticals; stock options from BlackThorn Therapeutics and an honorarium from Alkermes for activities unrelated to this project. No funding from these entities was used to support the current work, and all views expressed are solely those of the authors. The other authors declare no competing financial interests.

## Author summary

Many of the decisions we make involve weighing the costs and benefits of options in order to decide whether to approach or avoid an offer, such as deciding whether a new and advanced phone is worth the price. Major depressive disorder is associated with alterations in approach and avoidance behavior, but we know less about how the disorder is associated with solving the conflict of approaching or avoiding options with costs and benefits. Here we apply a computational model to investigate the cognitive mechanisms of solving this conflict, how these mechanisms are affected in depression, and how activity in brain regions involved in this process are informative for identifying the disorder. We found that depressed participants differed from healthy controls in both cognitive processes and in how brain activity was linked to these processes. Specifically, depression was associated with reduced sensitivity to benefits, but not costs (represented in the task by reward points and aversive images, respectively), a lack of bias to approach offers, and alterations in how the mapping of motor responses to approach or avoid offers influenced this bias. Further, we found that activity in nucleus accumbens and the pregenual anterior cingulate were informative in classifying disease status. Altogether, these findings indicate the utility in applying computational models to identify biomarkers of MDD in approach-avoidance conflict.

## Introduction

Adaptive decision making relies on using information in the environment to decide whether to approach or avoid stimuli, as when a predator chooses to approach or avoid a prey [1]. These decisions depend on the weighting of costs and benefits of approaching a stimulus (e.g., eating a mushroom that will increase satiety but might be toxic). In the example of foraging, too much approach (due to increased subjective value of reward or ignoring aversive outcomes) can be risky, while too much avoidance (due to decreased subjective value of reward or increased sensitivity to aversive outcomes) results in forgoing positive outcomes.

How individuals solve the conflict of whether to approach or avoid is of great interest to understanding behavior in mood disorders such as major depressive disorder (MDD), which is associated at the group level with both decreased approach behavior [2] and increased avoidance behavior [3,4]. However, ultimately, for cognitive or neural measures to be clinically useful, the field needs to go beyond group-level differences to making predictions about individuals. Here, we apply "computational multidimensional functional profiling" [5–7] to disentangle parameters underlying the dynamics of the decision process [8], and to leverage data-driven methods to assess whether a combination of such parameters is maximally predictive of relevant phenotypes and brain states [5–7]. Previous studies in movement disorders have shown that these methods can improve identification of relevant clinical variables, and that they can be superior to classification based on the raw data, or summary statistics thereof [6]. This approach therefore shows promise toward development of more effective, principled diagnostic and therapeutic strategies for mental illness.

Here, we decomposed the cognitive processes underlying approach-avoidance decision-making with an affective variant of the drift diffusion model (DDM) [9], a sequential sampling model often applied to two-choice decision making as an accumulation-to-bound process. Although originally used for perceptual or memory-based decisions, the DDM has been extended to capture value-based decisions, and their response time (RT) distributions, based on costs and benefits [10–13]. We further extend this model to capture approach-avoidance

decision-making in the presence of conflicting rewarding and aversive consequences (see [14] for a related approach). To do so, we consider various factors that could impact the underlying decision dynamics and could account for distinct forms of variability in MDD.

First and foremost, an instrumental decision to approach involves a larger weighting of the potential benefits over the potential costs of that decision. This relative weighting is thought to involve striatal mechanisms including the caudate nucleus [15], and, for approach-avoidance conflict, the pregenual anterior cingulate (pACC) [16–18]. Second, a Pavlovian bias could potentiate approach when the response needed to do so is congruent with approach tendencies (i.e., bringing a stimulus toward vs. away from oneself [19,20]). Such tendencies are related to nucleus accumbens and striatal dopamine mechanisms [21–23]. Third, according to sequential sampling models, a decision maker accumulates evidence to a bound, where the height of that bound determines the level of cautiousness and hence the speed-accuracy tradeoff [24]. The DDM allows us to assess both any starting point biases toward one bound or another (manifested in terms of changes in response proportions and fast RTs), but also how malleable the decision bound is. In particular, when decision conflict is experienced, the ACC and subthalamic nucleus (STN) are typically engaged to adjust the decision bound and to regulate impulsive choice [13,25,26].

MDD is a heterogeneous condition that may involve disturbances in any or all of the above processes. Indeed, MDD has been associated with decreased reward sensitivity [27] and altered neural responses in the caudate nucleus across several tasks [28,29]. MDD has also been linked to alterations in Pavlovian-Instrumental-Transfer, which captures the influence of background Pavlovian valence on instrumental decision making [30,31]. Finally, midcingulate responses to conflict and errors, which may be used in adjusting decision bounds, have also been linked to anxiety and depression [32].

Here, we quantitatively assessed the dynamic processes of approach-avoidance conflict decision-making in individuals with MDD and healthy controls, using hierarchical Bayesian parameter estimation of the DDM applied to behavioral and neuroimaging data described in [33]. We found that, as a group, MDD individuals exhibited (i) a reduced starting point bias to approach offers, (ii) a reduced reward sensitivity on evidence accumulation, and (iii) an opposite Pavlovian bias compared to controls. Moreover, these associations were further moderated by the differential impact of key neural signals on model parameters. In particular, MDD individuals exhibited trends for differences in the impact of pACC activity on evidence accumulation, and differential impact of nucleus accumbens on starting point bias. The combination of these computational biomarkers aided in classifying individual patient status and were associated with clinical measures in MDD. Finally, computational modeling of behavioral data collected during a follow-up session 6 months after the baseline session replicated the effect of reduced reward sensitivity in MDD, corroborating this effect as a promising computational biomarker of MDD.

## Methods

The current study utilized human data described in [33], which provides more details about data collection. Here, we describe the sample, the task, the methods used to extract trial-by-trial BOLD activation from regions of interest and the computational model fitted to data.

### Ethics statement

All participants gave written informed consent to a protocol approved by the Partners Human Research Committee.

## Participants

Twenty-one unmedicated female adult participants (mean age 25.2 ± 5.1 years) with Major Depressive Disorder (MDD) and 35 age-matched healthy female controls (mean age 26.3 ± 7.6 years) participated in the study. Six healthy control (HC) participants and one participant with MDD did not complete the study. Two additional MDD participants were excluded from analyses because their diagnosis was later found to be unreliable. Two HC participants were excluded because of a technical issue with registering their task responses. Three additional HC participants were excluded as their task performance was unreliable. The final sample included 18 participants diagnosed with MDD and 24 healthy controls. For more details about the sample, see S1 Text and [33]. A subset of 10 participants diagnosed with MDD and 17 healthy controls also performed the same task at 6-month follow-up and were administered several clinical scales.

## Task

Participants performed 105 trials of an approach-avoidance conflict task (Fig 1A) adapted from a prior non-human primate study [16]. For each trial, participants had to choose whether to approach or to avoid an offer. Approach decisions would lead to points, but also an aversive

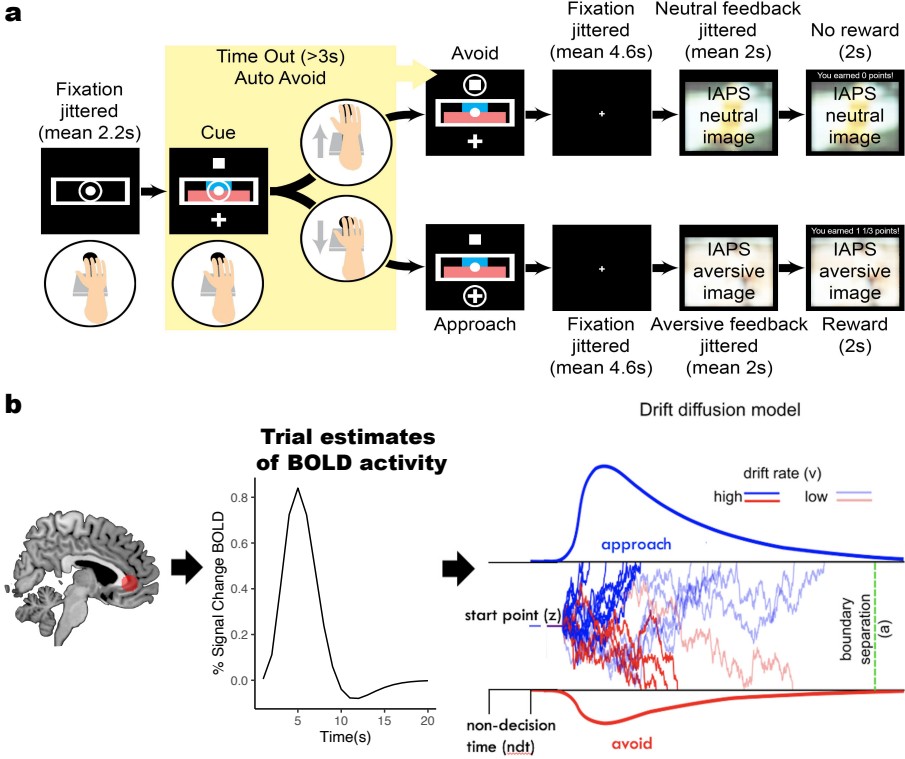

**Fig 1. Experimental task and computational model. a**, Participants used a joystick to decide whether to approach or avoid a combined offer of reward (points) and aversiveness (aversive stimuli). The magnitude of offered reward and aversiveness were represented by the width of the blue and red bar, respectively. **b**, Illustration of drift diffusion model applied to approach-avoidance conflict task. Trial-by-trial BOLD activity from regions of interest were used as regressors to measure their impact on decision parameters in the DDM. The DDM adapted to the approach-avoidance conflict task was used to estimate how trial-by-trial changes in offered reward and aversiveness (as well as neural correlates) altered the speed and sign of evidence accumulation (controlled by the drift rate (v)) to decide to approach (upper bound) or avoid (lower bound) offers. The boundary separation (a) parameter measures the distance between decision thresholds, the starting point bias (z) measures a priori tendencies to approach or avoid offers, while the non-decision time (ndt) parameter captures time spent on stimulus encoding and motor response. The illustration of the DDM is adapted from [41].

outcome (seeing an aversive picture accompanied by a matched aversive sound). Avoidance decisions resulted in no reward accompanied by the presentation a neutral image and neutral sound. Images were taken from the IAPS database [34]. The amount of points and the degree of aversiveness offered on each trial were parametrically varied and represented as the width of a blue (points) and a red bar (aversiveness of the image based on IAPS normative values). Sixteen levels of reward points and six levels of aversiveness of images, based on normative ratings from [34], were used. The value of both stimuli ranged from 0 to 5. Participants did not receive offers in which both stimuli had values of 0. Approach decisions were made by using a joystick to move a cursor to a plus sign, while avoidance decisions were made by moving the cursor to a square sign. The position of the response signs (above or below the bars) varied from trial to trial in a random and counterbalanced way. The task was separated into three runs with short breaks between runs. The entire task took approximately 15 minutes to complete. Six months after the baseline session, participants were invited to return to the laboratory for a follow-up session, in which the approach-avoidance task was re-administered and clinical symptoms were assessed.

## fMRI preprocessing and extraction of trial-by-trial activity

Functional MRI data were preprocessed and analyzed using Statistical Parametric Mapping software (SPM12; http://www.fil.ion.ucl.ac.uk/spm). Distortion correction was applied using field maps. Functional images were then realigned to the mean image of the series, corrected for motion and slice timing related artifacts, co-registered with the anatomical image, normalized to the 2 x 2 x 2 mm MNI template, and smoothed with an 8mm Gaussian kernel. We extracted trial-by-trial parameter estimates during the decision phase of regions of interest (ROIs) using a least squares separate (LS-S) approach [35], in which a separate trial-specific design matrix is used to obtain the activation estimate for each trial. In this approach the design matrices contain two regressors, one for the trial of interest plus a second that models all other trials simultaneously and additional covariates of no interest including motion realignment parameters and outliers calculated using Artifact Removal Tool in SPM [36]. For example, the activation estimate for trial 1 has a regressor modeling that trial and a second regressor modeling all other 104 trials. The estimate for β1 from this first design is the activation for trial 1. This process is repeated 105 times to obtain estimates for all trials. Based on prior findings in this area (e.g. [17], regions of interest included pregenual anterior cingulate cortex (defined as a single 12-mm sphere drawn around coordinates from a meta-analysis [37], caudate nucleus and nucleus accumbens (Oxford-Harvard subcortical atlas, 50% threshold) and subthalamic nucleus (FSL subthalamic nucleus atlas, 50% threshold).

## Behavioral analysis

Measures of response time and rate of approach across individuals with MDD and healthy controls were analyzed with linear and logistic regression models, respectively, using the BRMS package [38] created in STAN [39], a toolbox in R for doing Bayesian hierarchical estimation through Markov chain Monte Carlo (MCMC) sampling. MCMC sampling is a method for approaching the posterior distribution through sampling and can be used to estimate not only the mean and standard distributions of parameters from data fit to likelihood distributions, but also the uncertainty in these estimates, reflected in the width of the sampled posterior distribution. Further, we ran the models in separate chains (running the same model multiple times) and calculated the $\hat{R}$ convergence statistic [40], to verify that similar posterior distributions were approximated across chains. Finally, in hierarchical Bayesian analysis, estimation of group and subject parameters mutually informs each other using the group

distribution as a prior for the likelihood of individual estimates. This can improve estimates of individual parameters in models with few trials per subject [41].

## Modeling analysis

To quantify the dynamics of decision-making processes for approach-avoidance, we leveraged the drift diffusion model (DDM), a sequential sampling model that provides an algorithmic account of how evidence accumulation contributes to a binary decision process (Fig 1B) [9,24]. The DDM quantitatively captures the degree to which RT distributions and choices are accounted for by changes in latent decision parameters such as drift rate and decision threshold, which have orthogonal influences on accuracy and RT: higher drift captures greater information in the stimulus and results in shorter RT and better accuracy, whereas higher threshold captures increased response caution and results in longer RT and better accuracy. For preference-based decision-making tasks such as the approach-avoidance conflict task, in which neither decision is 'accurate', the decision threshold captures a tradeoff between speed and choice consistency (i.e., the tendency to make the same choice across trials with equal offers of reward and aversiveness) rather than speed-accuracy, while the drift rate captures the speed of evidence accumulation towards approaching or avoiding offers. The bias parameter captures the starting point of the accumulation process. Non-decision time (ndt) accounts for time spent on sensory encoding and motor response. The DDM is most commonly used to account for decision making in noisy sensory environments, but it has proved equally valuable for understanding dynamics of value-based decisions, whether during [12] or after value acquisition [13] or selected based on preference [42]. Here, we modeled motivated decisions by assuming that trial-by-trial values of reward and aversiveness drive evidence accumulation (captured by the drift rate parameter v) towards choosing to approach or avoid an offer (Fig 1, left). We adopted a Hierarchical Bayesian parameter estimation of the DDM using the HDDM-toolbox [41] to assess the impact of reward and aversiveness across MDD and healthy controls.

**Model comparison.** Model comparison was performed by beginning with a base model and systematically assessing whether adding a theoretically meaningful component improved model fit. The final model included all components that improved model fit. For drift rate, we assessed whether the impact of reward and aversiveness was of a linear or a logarithmic form, and whether there were additional effects of offers of 0 value (i.e., no reward or neutral aversive stimuli). For decision threshold, we tested whether it was modulated by decision conflict (measured as the absolute difference in values of offered reward and aversiveness [26]). Lastly, as a measure of Pavlovian bias, we tested whether approaching (avoiding) offers by pushing (pulling) the joystick to respond or vice-versa affected the starting point bias. The final model reported here included a log-transformation of reward, an impact of offers of 0 reward, the effect of conflict on decision threshold and Pavlovian bias on starting point bias (see Table A in S1 Text for model description and model fit metrics).

After establishing the best-fitting model to average behavior, we augmented the model to determine whether decision parameters are altered on a trial-by-trial basis as a function of BOLD activity in the pACC, caudate nucleus, nucleus accumbens and subthalamic nucleus, and to assess how this impact differed between groups. Estimating the impact of neural correlates on decision parameters quantifies how ROIs are linked to mechanistically meaningful parameters, over and above the effect of behavioral manipulation. Thus, trial-by-trial drift rate (V) was calculated as:

$$v_t \sim \log(\text{reward}_t) * \text{caudate nucleus}_t + \text{aversiveness}_t * \text{pACC}_t + \text{dReward}_t,$$

where reward and aversiveness represented the offered reward and offered aversiveness on

trial $t$, caudate nucleus and pACC were the activation during the decision phase on trial $t$. The pACC was hypothesized to be associated with aversiveness due to its causal role in increasing avoidance decisions in non-human primates [16], while the caudate nucleus is commonly associated with representing reward (e.g. [15]), and specifically found to correlate with reward during approach-avoidance conflict [43]. Offered reward was found to have a non-linear impact on drift rate, resulting in improved model-fit when offered reward was log-transformed, and when allowing drift rate to vary depending on whether offered reward was zero (dReward = 1) or non-zero (dReward = 0).

Decision threshold (a) was calculated as a baseline distance between decision boundaries and the impact of trial-by-trial conflict and activation in the STN, a region strongly implicated in adjusting decision threshold under conflict [25,26]:

$$a_t \sim |\text{reward}_{s,t} - \text{aversiveness}_t| * \text{subthalamic nucleus}_t$$

Starting point bias (Z) was calculated as a baseline starting point and the influence of Pavlovian bias and activity in nucleus accumbens:

$$z_t \sim \text{PavlovianBias}_t * \text{nucleus accumbens}_t$$

where 'PavlovianBias' was a dummy variable representing whether the mapping of response was push to approach and pull to avoid (PavlovianBias = 1) or vice-versa. The mapping changed from trial-to-trial, and participants used response cues (plus sign for approach and square sign for avoid) to figure out the mapping on each trial. The nucleus accumbens was assumed to impact starting point bias due to its association to Pavlovian biases reported in previous studies [21,23]. The model captured choice and RT for each trial (t) with the Wiener first passage time (wfpt) likelihood function of the DDM using the following calculation:

$$\text{choice} + \text{rt}_t \sim \text{wfpt}(a_t, \text{ ndt}, z_t, v_t),$$

Where ndt = non-decision time. For baseline/intercept-parameters we used priors from the HDDM package, which are informed by a wide range of empirical studies but are sufficiently conservative to allow for deviations in mean parameters based on the data [41]. Slope-coefficients used noninformative priors centered at 0. Intercept-parameters were estimated separately for each participant, while other coefficients were estimated on a group level, due to the inherent noise in neural signals. All predictor variables, with the exception of dummy-coded variables, were z-transformed prior to analysis.

**Model validation.**   We used Bayesian hierarchical estimation to fit the DDM to data. The models were run 5 times, each time with 5000 samples. The first 2500 samples were discarded as burn-in, i.e., to let the sampler identify the region of best fitting values in the parameter space. To capture potential differences between individuals with MDD and healthy controls we ran the model separately for the two groups, using the same prior distributions. We also modeled data from the two groups together in a mixed-effect regression model to directly estimate their group differences in BRMS, which gave nearly identical results (see Fig A in S1 Text for results from mixed-effect model).

The models were run 5 times to test whether for each so-called chain the model would converge on the same estimated parameter values. The models were deemed to have converged as the $\hat{R}$ statistic was below 1.1 for all parameters. The $\hat{R}$ statistic measures the degree of variation between chains relative to variation within chains [40]. This statistic will be close to 1 if the samples of the different chains are indistinguishable, and values below 1.1 are commonly deemed to identify a converged parameter.

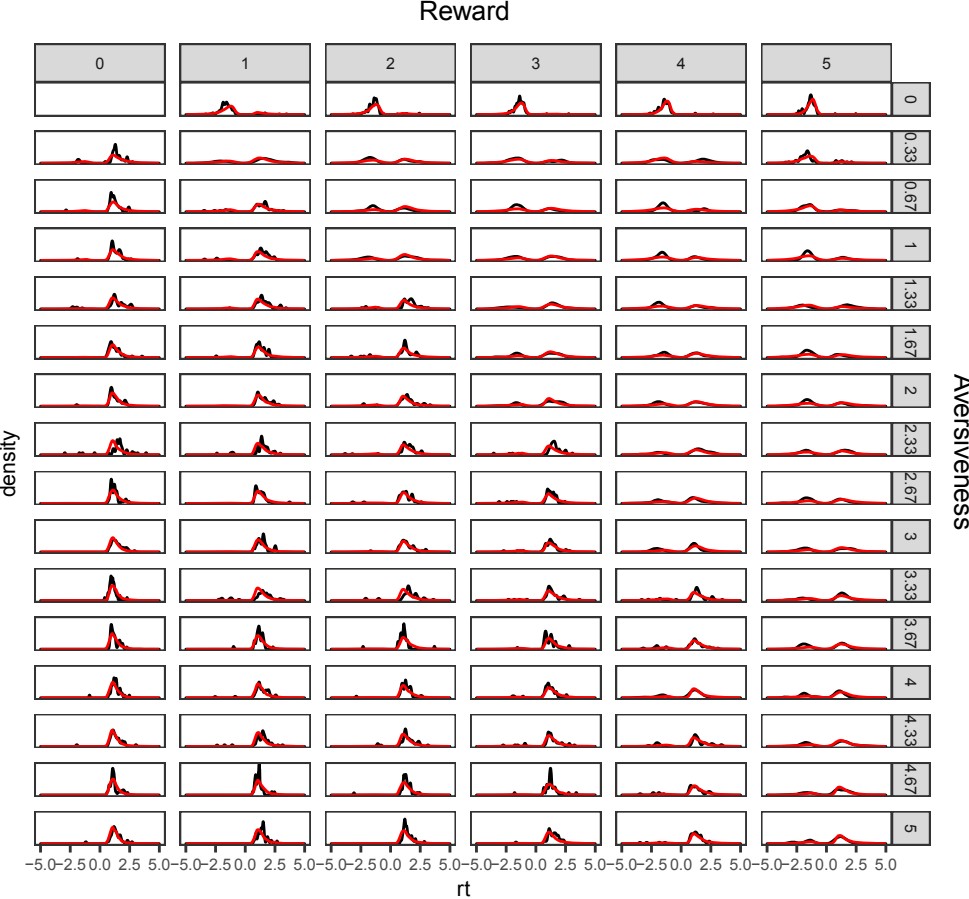

**Fig 2. Posterior predictive check.** The figure illustrates the models' (red lines) ability to recreate observed (black lines) choice and response time distributions across all combinations of offered reward and aversiveness. Avoidance decisions are set to have negative response times to distinguish the reaction time distribution of decisions to approach and avoid offers and to indicate the relative proportion of (observed and predicted) approach and avoidance decisions across combinations of reward and aversiveness.

The models' ability to capture choice and response time patterns was assessed by comparing observed and model-generated choice and response times (Fig 2). This posterior predictive check shows that the model captures changes in choice patterns and the distribution of response times across combinations of reward and aversiveness. See Fig B in S1 Text for posterior predictive checks for each subject.

In the Bayesian tradition we test effects as the posterior distribution of difference between group posterior distributions and report the probability of one group having a higher estimated parameter value as the proportion of the distribution of difference above 0 [44]. We report the 95% highest density interval (HDI) as uncertainty in the posterior distribution [44].

## Classification

To estimate any potential advantage of the computational modeling approach, we trained two classifiers on disorder status. One classifier used individual measures of brain activity (from ROIs) and behavioral results (RT and rate of approach), while the other used individually-estimated DDM parameters and their modulation by neural regressors. Importantly, these

parameters are estimated from a model that did not have access to clinical status (i.e., all subjects are estimated with a single group distribution), to prevent classification bias that could otherwise arise due to shrinkage (an effect in hierarchical Bayesian model estimation where individual parameters can be estimated closer to the group mean). A logistic regression classifier was trained 100 times using 10-fold cross-validation. The best-performing classifier from the training was then used to iteratively predict diagnosis status on 30% of held-out data. The performance of the classifier was measured on held-out data using the Area Under the Receiver-Operator-Curve (AUC) statistic, which can be interpreted to measure the probability of correctly choosing two randomly drawn samples from each the two classes (MDD and controls).

## Results

To investigate the mechanisms underlying approach-avoidance conflict decision-making in MDD, we applied a drift diffusion model to data from 18 adult females diagnosed with MDD and 24 psychiatrically healthy controls.

### Behavioral results

Overall rates of approach and response times across groups were analyzed with logistic and linear regression models, respectively. These analyses did not show an effect of MDD on either rate of approach (p(HC>MDD) = 0.606) or response time (p(HC>MDD) = 0.536).

### Computational modeling

Model comparison was performed by beginning with a base model and systematically assessing whether adding a theoretically meaningful component improved model fit. The final model reported here included a log-transformation of reward, an impact of offers of 0 reward, the effect of conflict on decision threshold and Pavlovian bias on starting point bias. We posited that if the ROIs of interest are related to approach/avoidance decision making as informed by prior literature on these regions, then taking into account their variability could improve estimation of decision parameters. We thus estimated how neural regressors impacted these processes, specifically estimating the influence of trial-by-trial variability in pACC and caudate nucleus on drift rate (ie. weighting of reward vs aversive attributes [16,17,33]), of STN on decision threshold [13,26,45,46] and of nucleus accumbens on starting point bias toward approach [21–23]. The best-fitting model was estimated to have converged as the $\hat{R}$ statistic was below 1.1 for all parameters [40], and was shown to recreate observed choice and RT patterns (Fig 2 and Fig B in S1 Text). See Methods for more details on model comparison and model validation.

### Drift rate

By capturing motivated approach-avoidance conflict decisions with the DDM, we assumed reward values would be accumulated as evidence for an approach response, whereas aversive values would contribute evidence for an avoidance response. These assumptions were confirmed; in both groups, trial-to-trial variations in reward were estimated to drive drift rate toward approach decisions while values of aversiveness influenced drift rate towards avoiding offers, indicated by coefficients that were credibly different from 0 (Fig 3 and Table 1). However, the impact of reward on drift rate was reduced in MDD compared to controls (p(HC>MDD) = 0.99). By contrast, sensitivity to changes in aversiveness did not differ between groups (p(HC>MDD) = 0.575) (Table 1). The intercept of drift rate did not differ between

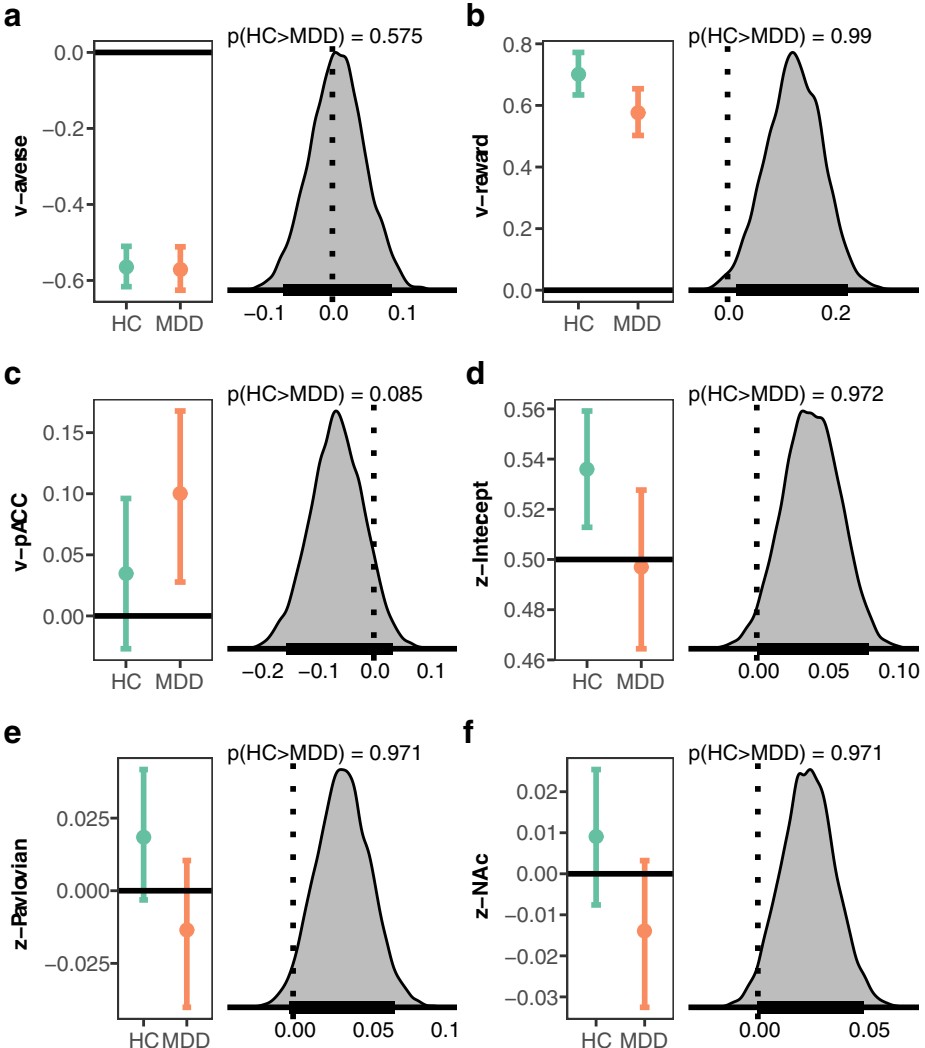

**Fig 3. Selected results from the computational model.** For each coefficient the left plot shows the group posterior distribution for healthy controls (HC) and individuals with major depressive disorder (MDD). The right plot shows the posterior distribution of difference as a measure of the effect of group on each coefficient, and the probability given data that the coefficient is higher in HC than MDD. **a**, weight of aversiveness onto drift rate (v), **b**, weight of reward onto drift rate (v), **c**, estimated relative starting point (z) between decision thresholds, **d**, impact of Pavlovian effect onto starting point (z), **e**, impact of activity in nucleus accumbens (NAcc) onto starting point (z), **f**, impact of activity in the pACC onto drift rate (v). For the entire set of coefficients see Table 1.

MDD and controls (p(HC>MDD) = 0.305), consistent with the behavioral results of similar rates of approach in MDD and HC.

While the trial-by-trial variations in reward and aversion affected drift rate in opposing directions, we hypothesized that trial-by-trial BOLD activity could serve as a proxy for motivational state and further modulate drift rate over and above the objective offered reward and aversion metrics. Accordingly, we modeled the impact of activity in the caudate nucleus and pACC onto drift rate with the a priori assumption that caudate nucleus would be associated with increased sensitivity to reward and pACC to aversiveness. Activation in caudate was not found to credibly influence drift rate, as the coefficient was estimated to overlap with 0 in both groups (Table 1), although it was estimated to be somewhat more positive in healthy controls

**Table 1. Posterior distributions of group parameters.** Lower and upper represent the lower and upper bound of the 95% highest density interval of the posterior distribution

| parameter | coefficient | HC | | | MDD | | | |
|---|---|---|---|---|---|---|---|---|
| | | mean | lower | upper | mean | lower | upper | p(HC>MDD) |
| threshold (a) | intercept | 2.244 | 2.083 | 2.430 | 2.259 | 1.991 | 2.550 | 0.467 |
| | STN | 0.055 | -0.031 | 0.134 | 0.058 | -0.025 | 0.138 | 0.484 |
| | conflict | 0.182 | 0.112 | 0.248 | 0.145 | 0.077 | 0.215 | 0.772 |
| | conflict:STN | 0.042 | -0.021 | 0.101 | 0.008 | -0.057 | 0.072 | 0.772 |
| non-decision time (ndt) | intercept | 0.737 | 0.656 | 0.818 | 0.661 | 0.561 | 0.760 | 0.882 |
| drift rate (v) | intercept | 0.665 | 0.409 | 0.942 | 0.783 | 0.399 | 1.160 | 0.305 |
| | reward | 0.701 | 0.634 | 0.772 | 0.577 | 0.502 | 0.654 | 0.990 |
| | aversiveness | -0.564 | -0.616 | -0.510 | -0.571 | -0.626 | -0.511 | 0.575 |
| | caudate | 0.012 | -0.050 | 0.073 | -0.035 | -0.102 | 0.036 | 0.838 |
| | pACC | 0.035 | -0.027 | 0.096 | 0.100 | 0.028 | 0.168 | 0.085 |
| | reward:caudate | -0.027 | -0.083 | 0.027 | 0.014 | -0.040 | 0.068 | 0.159 |
| | aversiveness:pACC | -0.006 | -0.060 | 0.046 | -0.017 | -0.078 | 0.041 | 0.605 |
| | Dreward | -1.105 | -1.299 | -0.918 | -0.974 | -1.191 | -0.776 | 0.180 |
| starting point bias (z) | intercept | 0.536 | 0.513 | 0.559 | 0.497 | 0.464 | 0.528 | 0.972 |
| | Pavlovian bias | 0.018 | -0.003 | 0.042 | -0.014 | -0.040 | 0.010 | 0.971 |
| | accumbens | 0.009 | -0.008 | 0.025 | -0.014 | -0.033 | 0.003 | 0.971 |
| | accumbens:Pavlovian bias | -0.001 | -0.025 | 0.022 | 0.011 | -0.013 | 0.034 | 0.242 |

STN, subthalamic nucleus; pACC, pregenual anterior cingulate cortex

(p(HC>MDD) = 0.838). Activity in pACC was associated with increased drift rate towards approach in MDD but not for controls, with a trending effect for more positive influence of pACC on drift rate in MDD (p(HC>MDD) = 0.085).

### Starting point bias

Changes in motivational state could influence changes in starting point, where a priori biases to approach or avoid offers (i.e., before seeing offered reward and aversiveness) would be represented, respectively, by a relative starting point toward the approach or avoid decision boundaries (Fig 1B). Individuals with MDD were not found to display a bias in either direction, as the starting point was estimated to be centered between approach and avoid decision boundaries (Fig 3 and Table 1). In contrast, healthy controls displayed a bias to approach, and this bias differed credibly from that in the MDD group (p(HC>MDD) = 0.972). We further measured the impact of response congruency on starting point, hypothesizing that a Pavlovian approach bias could be present when the mapping (which varied from trial to trial) was such that responses to approach were made by pushing the joystick to the response stimulus. Indeed, healthy controls were somewhat biased to approach when an approach-decision required a push of the joystick ($\beta_{Pavlovian}$(HC) = 0.018, HDI = - 0.003, 0.042). In contrast, this mapping moved the starting point in MDD further towards avoid ($\beta_{Pavlovian}$(MDD) = -0.014, HDI = -0.04, 0.01), and the effect between groups differed (p(HC>MDD) = 0.971).

Prior literature links nucleus accumbens activity to approach and avoidance of rewarding and punishing stimuli [21–23]. We thus also estimated the impact of variability in nucleus accumbens activity on starting point and its interaction with Pavlovian bias. Although the posterior distribution for both the MDD and control groups overlapped with zero, there was a reliable difference between the two groups (p(HC>MDD) = 0.971): increases in accumbens activity were related to a starting point bias towards approach in controls ($\beta_{nucleus\ accumbens}$(HC) = 0.009,

HDI = -0.008, 0.025) and towards avoidance in MDD ($\beta_{\text{nucleus accumbens}}$(MDD) = -0.014, HDI = -0.033, 0.003). These group effects are complemented by individual classification and clinical prediction below.

### Decision threshold

The distance between decision thresholds controls the amount of evidence needed to commit to a choice, and hence balances the tradeoff between speed and accuracy (or here, choice consistency, since accuracy is subjective). We expected that similar values of reward and aversion could elicit conflict and induce the need to accumulate more evidence (higher decision threshold) before committing to a choice. Speed-accuracy tradeoffs, measured by the width of decision thresholds, did not differ between groups (p(HC>MDD) = 0.467), nor did the impact of conflict onto this tradeoff (p(HC>MDD) = 0.772). Based on previous findings that the mid-cingulate can signal to STN the need to accumulate more evidence via an elevation in decision threshold [13,26,45,46], we estimated the impact of STN on decision threshold and its interaction with decision conflict, i.e., when values of offered reward and aversiveness were of similar value. Although activity in STN was associated with somewhat increased decision threshold, the association of STN and threshold did not differ between MDD and controls (p(HC>MDD) = 0.484). Further, there were no effects of the interaction between STN and conflict onto decision threshold within ($\beta_{\text{STN:conflict}}$(HC) = 0.042, HDI = -0.021, 0.101), ($\beta_{\text{STN:conflict}}$(MDD) = -0.008, HDI = -0.057, 0.072) or between groups (p(HC>MDD) = 0.772).

### Classification of clinical status based on computational biomarkers

As noted above, DDM parameters reliably differed between HC and MDD groups, despite few observable differences in the average behavioral psychophysical functions. Ultimately, however, we are interested in the utility of computational markers for making inferences about individuals, rather than groups as a whole. We thus fit a single hierarchical DDM model across both populations (so that we did not bias individual estimates to be similar for each group; see Methods) and extracted individual subject posterior distributions. We then built a classifier using regularized logistic regression and cross-validation to predict clinical status based on individual parameter estimates, testing the classifier on held-out data. We repeated this same data-driven procedure but using only behavioral variables and brain correlates thereof, without model parameters. As shown in Fig 4, this procedure was moderately successful in improving sensitivity and specificity of MDD predictions, but only when model parameters were used (AUC = 0.68). Indeed, classification without computational biomarkers did not exceed chance (AUC = 0.47). Moreover, a classifier using computational DDM parameters but omitting the biomarkers (neural regressors) performed at an intermediate level (AUC = 0.58). (The AUC statistic can be interpreted to measure the probability of correctly classifying two randomly drawn samples from each the two classes, or alternatively, it is the true positive rate averaged across all possible values of false positives.) Altogether, this finding demonstrates the potential utility of computational biomarkers for classification.

We next assessed which parameters were most diagnostic. Interestingly, the impact of nucleus accumbens onto starting point was estimated to be the most distinguishing feature for predicting disorder status (Fig 4). Recall that at the group level, nucleus accumbens activity was oppositely predictive of starting point biases toward approach vs avoidance in HC vs MDD. The finding that this feature is the most diagnostic for distinguishing patients from controls at the individual level suggests that it is reliable and not dependent on outlier participants. Similarly, the influence of pACC on drift rate was also a distinguishing feature. Other important parameters include the overall starting point bias, the Pavlovian effect, and the impact of

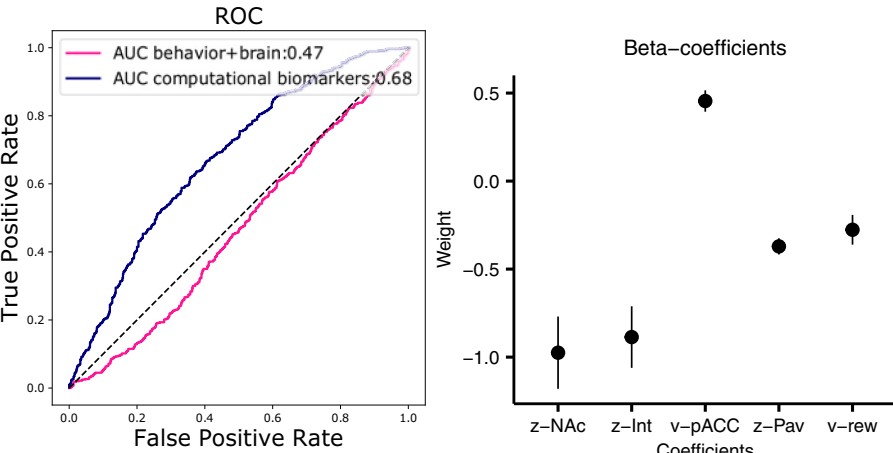

**Fig 4. Classification of MDD status and feature importance for the computational biomarker classifier. Left**, the receiver operating characteristic (ROC) curve and the ROC area under curve (AUC) statistic for a classifier using individual parameter values from the computational model (purple) and a classifier using mean observed behavioral measures of response time and approach rate and mean activity in ROIs (caudate nucleus, nucleus accumbens (NAcc), pregenual anterior cingulate cortex (pACC), and subthalamic nucleus (STN)). **Right**, mean estimated beta-coefficients from classifier with 95% confidence intervals for the classifier using computational biomarkers (purple). Coefficients are sorted by weight from left to right as the absolute distance from 0, the magnitude of which indicates the importance of each feature for the classification. Int = Intercept, PavBias = Pavlovian bias, v = drift rate, z = starting-point bias.

reward on drift rate, all consistent with findings at the group level, and thereby showing the utility of both brain and behavioral computational biomarkers of dynamic decision processes for clinical prediction.

## Clinical measures

We next evaluated how robustly these computational biomarkers relate to symptoms and prospectively predict disease course. We ran a multivariate multiple regression linking clinical measures collected at time of testing and 6-month follow-up (see Table C in S1 Text for full model definitions), within the MDD group. This analysis demonstrated that reward sensitivity (*v*-reward) was negatively associated with perceived stress ($b$ = -14.74 (CI = -28.57 –-0.90), $t(7)$ = -3.04, $p$ = .039), and the individual impact of the nucleus accumbens on starting point was associated with 6-month follow-up scores on the Hamilton Depression Rating Scale ($b$ = -1495 (CI = -2986.47 –-4.04), $t(8)$ = -2.4, $p$ = .04). These findings further reify the utility of the computational biomarkers for predicting symptom progression.

## Follow-up data

Six months after the original study, 10 participants with MDD and 17 healthy controls returned to the laboratory for a follow-up session. Three of the participants in the MDD group no longer fulfilled criteria for MDD, measured as symptom scores of 7 or lower on the Hamilton Depression Rating Scale [47], resulting in a significant overall reduction in symptom scores from the first to second session ($t(9)$ = 2.458, p = 0.036). Without these three participants the difference between session was no longer significantly different ($t(6)$ = 1.2914, p = 0.2441). However, due to the low sample size we chose to not exclude these participants. We applied the same computational model (without neural regressors) to this follow-up session.

**Test-retest reliability of model parameters and replication of reward sensitivity effect.**
We first assessed the reliability of individual parameter estimates by calculating two-way intra-class correlation coefficients of mean parameter estimates across the two tests. The intercept drift rate, sensitivities to reward and aversiveness onto drift rate and non-decision time parameters were significantly correlated across the two data collection phases ($p < 0.05$), while the remaining parameters were not ($p > 0.05$) (see S1 Text for statistics for each parameter). We found that the parameter estimates for starting point bias and decision threshold depended on the inclusion of neural predictors, thus providing a potential answer as to why these parameters did not correlate between time-points.

As can be seen in Fig 5 and Table B in S1 Text, replicating the primary model result, participants with MDD were also found to be less sensitive to reward in the follow-up data (p (HC>MDD) = 0.99). However, in contrast to the indistinguishable sensitivity to aversiveness in the original data, at follow-up, participants with MDD were found to be less sensitive to aversiveness (p(HC>MDD) = 0). The other parameters generally showed the same qualitative patterns, with the exception that the MDD group at follow-up were found to be biased towards approach (see Fig 5 for comparison of all parameters in the two datasets).

**Classification.**   To evaluate the generalization of the classifier distinguishing MDD from HC based on model parameters alone, we tested how well this same classifier (but without the neural regressors) could distinguish MDD from HC based on model parameters at follow-up. Because we had found at baseline that the neural regressors (biomarkers) were helpful for classifying patient status, we expected only moderate success in classifying the follow-up unseen data without such regressors. Nevertheless, we found that the classifier at follow-up predicted diagnosis with an accuracy of 62% (AUC = 0.55). Moreover, considering that three MDD subjects were considered in remission (and one was borderline), this classifier success improved to 69/73% (AUC = 0.59/0.61) if these subjects are considered to be healthy (HC) at follow-up. This latter result suggests that the classifier was not simply overfitting to individuals given that the same participants had different clinical status. Nevertheless, to further test whether the performance of this classifier was driven by including the same participant in the trained and tested sample, we also conducted as a purely out-of-sample test in which we iteratively predicted an individual at follow-up that was excluded during the training of the classifier. This approach led to reduced performance (59% accuracy), but was still found to improve when the three (and one borderline) remitters were considered to be healthy controls at follow-up, with an accuracy of 64% (69%).

## Discussion

We applied a computational model to data from an approach-avoidance conflict task in order to investigate the mechanisms of how individuals with MDD solve the problem of approaching or avoiding offers of combined reward and aversiveness. We found that individuals with MDD were less sensitive to changes in offered reward but did not differ from healthy controls on sensitivity to aversiveness. Controls were found to have an a priori starting point bias towards approaching offers, whereas the MDD group did not display such a bias. We also found that activity in the nucleus accumbens was associated with (trending) opposite influence on bias across groups, such that it led to greater approach bias in controls but greater avoid bias in MDD. Further, we found that Pavlovian congruency of the response mapping influenced starting point differently in the two groups, where pushing the joystick to approach offers lead to increased approach-bias in controls and increased avoidance-bias in MDD. Moreover, we showed that computational modeling improved classification of disorder compared to a classifier using raw behavioral and neural measures. Further, and highlighting

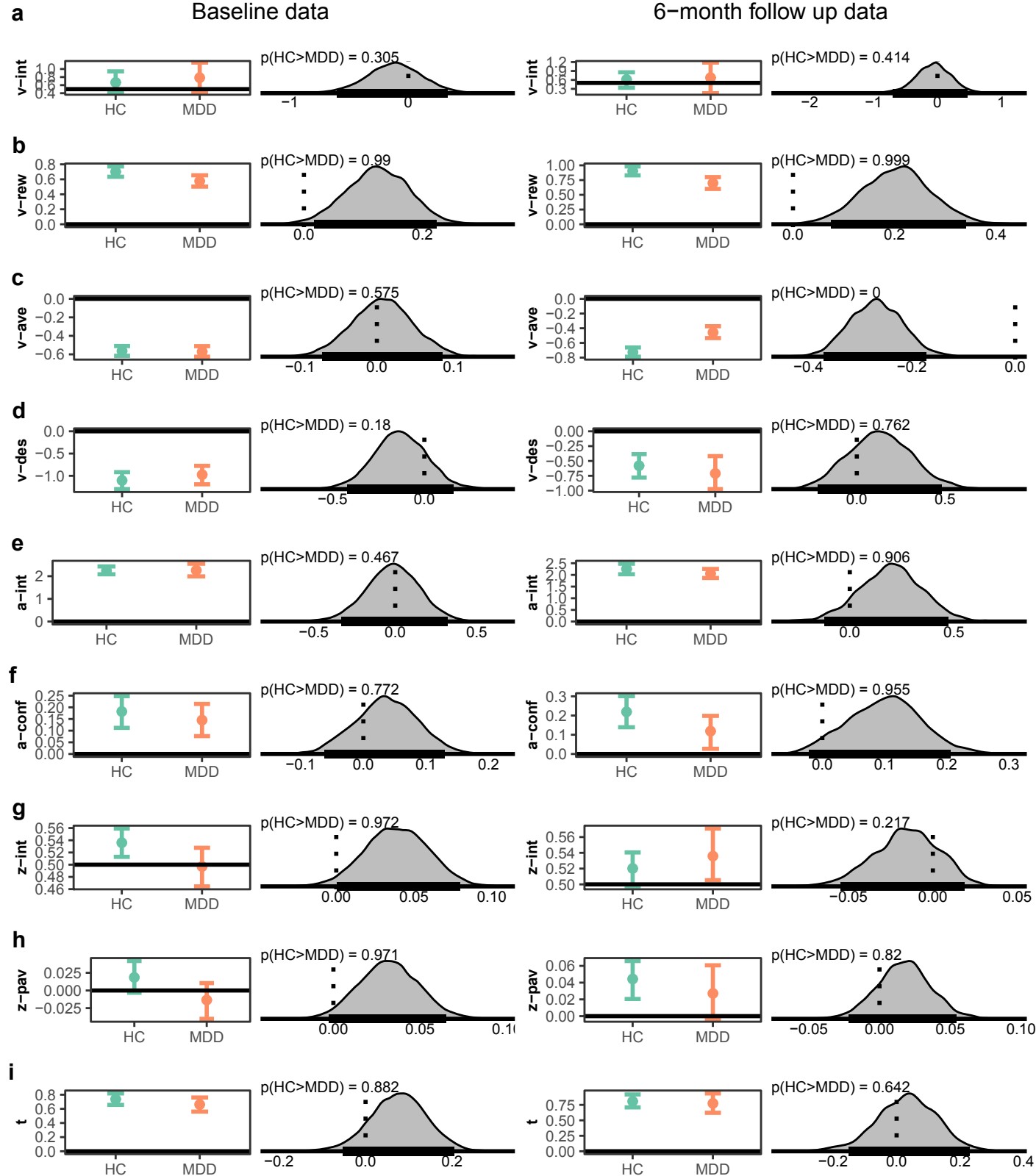

**Fig 5. Posterior distributions across MDD and HC at the original data collection and for the same task applied to a subset of participants at 6-month follow-up.** For each coefficient the left plot shows the group posterior distribution for healthy controls (HC) and individuals with major depressive disorder (MDD). The right plot shows the posterior distribution of difference as a measure of the effect of group on each coefficient, and the probability given data that the coefficient is higher in HC than MDD. **a**, Intercept drift rate (v), **b**, weight of reward onto drift rate (v), **c**, weight of aversiveness onto drift rate (v), **d**, dummy coding of reward onto drift rate (v) (1 = reward offer of 0, 0 = non-zero reward offer), **e**, Intercept value decision threshold (a), **f**, influence of conflict on decision threshold (a) **g**, estimated relative starting point (z) between decision thresholds, **h**, impact of Pavlovian effect onto starting point (z) and **i**, non-decision time (t), For statistics from posterior distributions in follow-up data see Table B in S1 Text. HC = healthy controls, MDD = Major Depressive Disorder.

incremental predictive validity, individual model coefficients were related to clinical symptoms of MDD at time of testing and predicted symptoms at 6-month follow-up. Finally, by analyzing data collected at 6-month follow-up we showed that individual parameters were stable across time and that the effects of reduced reward sensitivity in MDD was robust.

Cognitive process models, such as the DDM, offer an insight into the cognitive mechanisms underlying behavior, and can also be used to link hypotheses to neural mechanisms [26,48–50]. Understanding how these mechanisms are altered during approach-avoidance conflict in MDD thus has the potential of identifying computational biomarkers, which further can help bridge understanding of the implication of MDD on approach-avoidance conflict on behavioral and neural levels, provide a common framework for data from rodents [18], non-human primates [16,17,51] and humans [33], and potentially aid in individually tailoring treatment to patients. Future studies could also estimate the utility of computational modeling of approach-avoidance conflict in other psychiatric disorders, as abnormal approach-avoidance decision making have been implicated in, among others, anxiety disorders [52,53], eating disorders, substance use disorders, and personality disorder (see [54] for a recent review).

In contrast to our hypothesis, and self-report measures of approach and avoidance [2–4], there was not an overall reduction in the rate of approached offers in MDD [33]. However, individuals with MDD were less sensitive to changes in offered reward on drift rate (which is manifest in terms of both choice and RT) in both the original and follow-up datasets. This reduced reward sensitivity resulted in individuals with MDD accruing somewhat less reward points during the experiment (p(HC>MDD) = 0.873). A reduction in accrued points shows how insensitivity to reward can result in less positive outcomes. Reduction in reward sensitivity in MDD has been found in other tasks, including instrumental learning [27], and directly maps onto anhedonia, an important endophenotype of depression [55]. However, in the current study the measure of reward sensitivity was not significantly related to Snaith Hamilton Pleasure Scale [56], a self-report measure of anhedonia.

We also reported differences in a priori biases towards approaching offers, before any evidence of an individual trial's offer can be weighted. Whereas individuals with MDD did not display a bias to either approach or avoid (captured as a starting point equidistant between the two decision thresholds (Fig 1B)), the controls exhibited a starting point bias to approach. This effect resembles a lack of optimism bias observed in MDD [57]. However, the results from the follow-up study did not replicate the differences in starting point bias. A larger sample size could reveal whether these differences are the effect of having previously performed the task or reflect that MDD indeed are not associated with an altered a priori bias.

Intriguingly, we also found that variability in activity in the nucleus accumbens was associated with opposing effects on starting point biases in MDD and controls. One possible interpretation of this effect is that the nucleus accumbens is often characterized as reflecting the net subjective valuation of an individual–what the person "cares about" in sum–after accounting for various cognitive and affective influences [58,59]. MDD patients may have an altered nucleus accumbens subjective valuation that is biased toward avoidance and aversiveness, and hence activity in this region is more likely to induce a bias to avoid. Indeed, this effect was the

most predictive of disorder status at the individual level and predicted depression scores at 6-month follow-up. The opposite impacts of nucleus accumbens on approach and avoid tendencies in MDD vs HC may also relate to findings showing opposite effects of Pavlovian to Instrumental transfer in MDD [31] (but see [30]). In contrast, a recent study found that MDD did not differ from controls in tendencies to approach rewards and avoid losses in a go/no-go instrumental learning task [60]. Future studies could investigate the influence of Pavlovian biases and the interaction of learning vs. preference-based choices in MDD.

The classifier predicted diagnosis only when computational biomarkers were included. When applied to follow-up data without neural regressors, there was only a hint for classification in the right direction (62% accuracy, AUC = 0.55), partly reaffirming the need for neural regressors for more reliable classification. More optimistically, this classifier performance was more favorable (69% accuracy, AUC = 0.59) if we considered the remitters to be HC at follow-up, which would fit if the DDM parameters reflect a state rather than a trait. However, much larger sample sizes are needed to test this notion.

## Limitations

We found that individually estimated model parameters were related to clinical measures in individuals with MDD and predicted future status. However, larger samples are needed to confirm these promising results of identifying computational biomarkers of approach-avoidance conflict decision making in MDD. In addition, despite showing that the classifier with computational model parameters outperformed the classifier using behavioral variables and neural correlates, future studies should increase sample size to more reliably estimate the utility of using model parameters to classify MDD status. A larger sample size could also allow testing linear classifiers, using dimensional measures of mood, in contrast to the categorical outcome of healthy vs. MDD used here. Such an approach could describe to which degree whether the group effects observed here reflect effects of state or trait. The results of improved classification at follow up when categorizing remitters as healthy controls hints that it is capturing state. An increased sample size would also more conclusively reveal whether clinical measures of mood not found to be significantly associated with decision parameters in the current study indeed aren't associated, or whether these results reflect low power to detect such effects. Further, the dataset used here consisted of female participants, thus precluding us from evaluating the generalizability of the findings. We also note that the retention rate and, within MDD, rate of remitters, were somewhat low. This could reflect an issue of self-selection, where remitted participants were more likely to not come in for a second testing. Finally, alternative models and neural circuits beyond those we tested could be shown to provide a better fit to data. Given the sample size we focused only on theoretically motivated ROIs in the computational model.

## Conclusions

Computational modeling revealed that participants with MDD solved approach-avoidance conflict differently than healthy controls. In particular, MDD was associated with reduced reward sensitivity. Individual parameters were linked to clinical measures of MDD and were useful for classifying diagnosis. Collectively, these findings establish the promise of computational psychiatry approaches to dissecting approach-avoidance decision dynamics relevant for affective disorders.

## Supporting information

**S1 Text. Fig A.** Selected results from the computational mixed-effect model. For each coefficient the left plot shows the group posterior distribution for healthy controls (HC) and

individuals with major depressive disorder (MDD). The right plot shows the posterior distribution of difference as a measure of the effect of group on each coefficient, and the probability given data that the coefficient is higher in HC than MDD. **A**, weight of aversiveness onto drift rate (v), **B**, weight of reward onto drift rate (v), **C**, estimated relative starting point (z) between decision thresholds, **D**, impact of Pavlovian effect onto starting point (z), **E**, impact of activity in nucleus accumbens (NAcc) onto starting point (z), **F**, impact of activity in the pACC onto drift rate (v). The results from the mixed-effect model here overlap with the results from the model in Fig 3, in which the two groups were estimated separately. **Table A.** Description and fit of tested models. Model comparison was performed by comparing a baseline model to a model in which one 'component' was modified. The model we report from includes all the 'components' that improved fit compared to the baseline model. The function of the impact of reward and aversiveness onto drift rate was assumed to be linear or logarithmic, and were assessed on whether model fit was improved when including a dummy coded variable that indicated whether the offered value of reward (Dreward) or aversiveness (Daverse) was 0 (D = 1) or not (D = 0). Conflict was measured as the absolute difference in reward and aversiveness and was estimated to influence the decision threshold parameter. PavlovianBias included information on whether approaching (avoiding) offers involved pushing (pulling) the joystick to respond (PavlovianBias = 1) or vice-versa (PavlovianBias = 0). Lower values of DIC indicate better fit to data. DIC = deviance information criterion. **Fig B.** Observed (black) and predicted (red) response time distributions across subjects. Avoid-decisions are set to be negative to separate RT distributions for decisions to approach and avoid. **Table B.** Posterior distributions for group parameters at follow-up. Lower and upper represent the lower and upper bound of the 95% highest density interval of the posterior distribution. For comparison to results from the original dataset, the rightmost column represents probabilities of group difference from original dataset. **Table C.** Multivariate regression for association between clinical measures collected at time of testing and at 6-month follow-up to decision parameters. **Table D.** Intraclass correlation coefficient for individual parameters across sessions. (DOCX)

## Acknowledgments

The content is solely the responsibility of the authors and does not necessarily represent the official views of the funding agencies.

## Author Contributions

**Conceptualization:** Maria Ironside, Ken-ichi Amemori, Ann M. Graybiel, Diego A. Pizzagalli.

**Data curation:** Maria Ironside, Callie L. McGrath, Min S. Kang, Diego A. Pizzagalli.

**Formal analysis:** Mads L. Pedersen, Maria Ironside, Michael J. Frank.

**Funding acquisition:** Diego A. Pizzagalli.

**Investigation:** Maria Ironside, Diego A. Pizzagalli, Michael J. Frank.

**Methodology:** Mads L. Pedersen, Maria Ironside, Michael J. Frank.

**Project administration:** Diego A. Pizzagalli, Michael J. Frank.

**Resources:** Michael J. Frank.

**Software:** Michael J. Frank.

**Supervision:** Diego A. Pizzagalli, Michael J. Frank.

**Visualization:** Mads L. Pedersen.

**Writing – original draft:** Mads L. Pedersen, Michael J. Frank.

**Writing – review & editing:** Maria Ironside, Ken-ichi Amemori, Ann M. Graybiel, Diego A. Pizzagalli.

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
