## [Decision Letter · Decision Letter 0]

16 Jun 2020

Dear Mr. Pedersen,

Thank you very much for submitting your manuscript "Computational phenotyping of brain-behavior dynamics underlying approach-avoidance conflict in major depressive disorder" for consideration at PLOS Computational Biology.

As with all papers reviewed by the journal, your manuscript was reviewed by members of the editorial board and by several independent reviewers. In light of the reviews (below this email), we would like to invite the resubmission of a significantly-revised version that takes into account the reviewers' comments.

We cannot make any decision about publication until we have seen the revised manuscript and your response to the reviewers' comments. Your revised manuscript is also likely to be sent to reviewers for further evaluation.

Sincerely,

Jean Daunizeau

Associate Editor

PLOS Computational Biology

Daniele Marinazzo

Deputy Editor

PLOS Computational Biology

Reviewer's Responses to Questions

**Comments to the Authors:**

Reviewer #1: This interesting manuscript uses a drift-diffusion model where both aspects of stimulus value and BOLD activation in two brain regions, the caudate nucleus and pACC, are used to determine drift rates. In addition, the decision threshold multiplicatively depends on subthalamic activity and the absolute difference between rewarding and appetivive value aspects of stimuli, while Pavlovian bias multiplies nucleus accumbens activity to determine Pavlovian bias, here mapped to the starting point of the diffusion process. The model is used to characterise two samples, of 18 depressed, unmedicated (MDD) and 25 health control (HC) participants. The authors find that using both computational parameters and brain measures, but most importantly the former, they can classify the 1/3 held-out sample they use with an AOC of 68%. They also find that MDD participants differ from HC with respect to some computational parameters esp. the weight of reward (but not aversiveness) on the drift rate.

I hesitated to recommend major revision rather than rejection of this study because the very small sample of 18 MDD participants really does not allow conclusions to be drawn about MDD itself. However I felt that the methodology is intersting and the results should be published, so that they can be aggregated with similar studies and conclusions be drawn about this interesting group of disorders. This is especially important as MDD is likely to be a heterogenous group of disorders, and therefore small subsamples from this population may differ substantially from each other. The authors do recognise that the sample is small, but in my view this is a very important limitation. Really, the sample size is so small that holding out 1/3 means that predictive testing was only done on 6 depressed participants (and a proportionate number of controls). Therefore the 68% AOC should be regarded as a proof-of-principle analysis. Really, there is no grounds at all to suggest that the results point to clinical utility.

I am aware of how difficult it is to recruit unmedicated patients with major depressive disorder that don't meet exclusion criteria of all sorts in a practical length of time, for a single laboratory. This is why I do not recommend rejection of the paper. However, as the most predictive aspects of the modelling are the computational parameters, and hence recruiting a second lot of 25+25 participants for behaviour-only analysis with this one task may be feasible in a relatively short time, maybe tagging the task on MDD-HC studies.

The authors mention computational model comparison way too briefly in the manuscript, and even refer to another paper as details of the sample. As a reviewer, I found this very frustrating. Essentially, if the paper was to be published like this it would say to readers 'trust our reviewers, they went into the trouble to find out if the winning model really is better, if the excluded participants are likely to bias the results, etc.'. I think there should be clear evidence about model comparison in the results, good information about the sample and exclusions in the supplement, and discussion of both of these in the Discussion.

Reviewer #2: -This is a well-motivated and well-written paper, reporting results of sophisticated computational and Bayesian analyses of potential clinical relevance. While I am enthusiastic about the study, the paper has some limitations that should be addressed before publication – mainly having to do with clarity of methods/results. I expand on these issues below (many of which are minor), mainly in order of presentation in the text. I should note that several appear to relate to the fact that the methods were moved to the end, and the authors failed to define/explain things the first time they appear in the manuscript as it is currently organized.

-Around line 97 – The authors should here define what “reward sensitivity” means (as they use it in the paper) before using it below in a way that assumes the reader knows what it refers to.

-The acronym RT should be defined on first use.

-The authors mention pavlovian-instrumental transfer without explaining what it refers to, which is necessary for understanding how it connects to their own work as the authors mention.

-There are some potential (minor) issues with consistent terminology that may cause the reader some unnecessary confusion. For example, it appears the authors use “evidence accumulation” and “drift rate” as synonymous in certain places, where sticking with consistent terminology would be clearer (e.g., perhaps initially discuss how drift rate controls the speed of evidence accumulation, but then always refer to the actual drift rate parameter after that point).

-The authors state that their measures were “sufficient to classify individual patient status”. Without qualification (and/or stating how well classification performed above chance), this read overly strong to me.

-Figure 1 – The parameter variables should be added to the legend. At the moment, the reader can’t easily map the variables in the figure (e.g., z) to their names in the legend. Also, all parameters are illustrated except for non-decision time. It’d be nice if that could be included in the DDM illustration for completeness.

-Around line 160 - The authors assume a lot of reader knowledge about modeling methods. They should say something more about their model estimation and comparison methods (e.g., MCMC, what chains are, etc.). Some of this also isn't thoroughly explained in the methods at the end either.

-They provide little task detail, and not much more in the methods. More details are needed, and the reader shouldn’t need to go digging in previous manuscripts to understand basic aspects of the task (e.g., how many trials? What was stimulus aversiveness based on? How many different stimulus aversiveness and reward value combinations were used, and how many times each? How exactly was push-pull and approach-avoid orthogonalized in terms of trial order? etc.).

-Figure 3 – The legend refers to panels A, B, C, etc. However, these labels are nowhere in the figure. Also, the Y axes are unclear, both in terms of units and labels. Using the parameter names (e.g., “drift rate”) would be clearer, and/or they should state what the variables (e.g., z) refer to in the legend. In general, I felt like many figure details were not explained in the legend.

-Line 239 – It seems like a citation for this statement is missing.

-Discussion of the STN seems to come out of nowhere in the results, and the acronym is never defined. Discussion of its relevance should be in the introduction.

-The authors repeatedly mention a plausible role for the midcingulate. It made me wonder why this wasn’t included as an ROI.

-The acronym AUC is never defined and used only once in the text.

-The authors report AUC for the model with poor classification (showing it performs at chance levels), but do not show the AUC supporting above chance performance for the biomarker model. Both should be reported for consistency.

-Figure 4 and related text – The authors could help the unfamiliar reader understand some of this better. For example, do their reported AUC values indicate high classification accuracy? Or above chance but still fairly low? Some more intuitive indication of “how much” above chance their model did would be helpful. Similarly, the authors do not help the reader to understand how to interpret the beta values in this context. I think many readers will walk away from this figure (and associated text) fairly unclear about exactly how good classification was. They also need to define figure acronyms in the legend.

-The findings with clinical measures were often only borderline significant. It’d be nice to have a sense of the effect size here. Also, why not stick with the Bayesian approach here? Can you show something like a Bayes factor score that provides clear evidence for these effects?

-Around line 341 – They mention their analyses could be extended to anxious populations. It made me wonder why only to anxiety? Is this not potentially relevant for other psychiatric disorders as well?

-Around line 349 – The authors mention direct links to anhedonia. It made me wonder about possible ways to test whether their measures correlate with standard experimental measures of anhedonia (e.g., SHAPS). If the authors expected this connection with their model, why not include such measures?

-Line 363 – there is a missing word here.

-Throughout the discussion, I felt like there was a lot of re-stating results and fairly little actual interpretation. Do the authors not have some proposal about what their results mean (either mechanistically or clinically)? For example, how should we think about nucleus accumbens having opposite influences in each group? Or what might it mean that there is a reversed pavlovian effect?

-The limitations section discusses things that aren't limitations. It mainly talks about results they expected to find but didn’t, which says nothing about limitations in their methods directly. They also don’t mention limitations they could. For example, the sample size is quite small. Do power limitations prevent interpretation of false negatives? Also, the sample was female – which prevents generalizing these results to any depressed men.

-It felt like a conclusions section was missing after the limitations.

-Their methods refer the reader to other papers a lot, where understanding the study requires many of these details (especially in relation to the task, as mentioned above).

-I felt like the authors could expand on how to think about the influence of neural activity on parameters. It’s a bit complex, because these reflect different levels of description – where one assumes that computational mechanisms/parameters are implemented by (i.e., correspond to) something within neural hardware. But here it’s not that a given ROI is the thing that implements a decision parameter, but that it influences it in some way. So should this be thought of as that ROI affecting a different brain region/process? Or how should this be thought about?

-Around line 449 – They state they used priors informed by previous work. But what were they?

-I was a bit unclear about their description of model comparison. It says they added one component and then compared to baseline for each possible component. But components can interact. So did they not compare models with different combinations of included components?

-In summary, all of these points should be addressable and most reflect issues of clarity/completeness. I hope the authors find them helpful in improving the manuscript.

**Have all data underlying the figures and results presented in the manuscript been provided?**

Reviewer #1: No: The authors say that detailed data cannot be provided because of their sensitive clinical nature.

Reviewer #2: No: No data was provided.

PLOS authors have the option to publish the peer review history of their article (what does this mean?). If published, this will include your full peer review and any attached files.

Reviewer #1: Yes: Dr. Michael Moutoussis

Reviewer #2: No
---

## [Decision Letter · Decision Letter 1]

26 Oct 2020

Dear Mr. Pedersen,

Thank you very much for submitting your manuscript "Computational phenotyping of brain-behavior dynamics underlying approach-avoidance conflict in major depressive disorder" for consideration at PLOS Computational Biology.

As with all papers reviewed by the journal, your manuscript was reviewed by members of the editorial board and by two independent reviewers. In light of the reviews (below this email), we would like to invite the resubmission of a significantly-revised version that takes into account the reviewers' comments.

***

As you will see, reviewer #2 is happy about your revision, but reviewer #1 still raises a few methodological points that, I believe, need to be addressed. In particular, his first point requires a re-analysis of your data, using a proper use of hierarchical random-effect analysis (I fully agree wih his comment, which essentially states that the way you use hierarchical group-level inference may amplify trivial between-group differences because of the prior shrinkage effect around the group means). Although the other points may not require a re-analysis of the data, addressing them still requires a modification of the results and/or discussion sections. This is why my decision is still "major revision" at this stage. I hope you understand this decision.

*** 

We cannot make any decision about publication until we have seen the revised manuscript and your response to the reviewers' comments. Your revised manuscript is also likely to be sent to reviewers for further evaluation.

Sincerely,

Jean Daunizeau

Associate Editor

PLOS Computational Biology

Daniele Marinazzo

Deputy Editor

PLOS Computational Biology

Reviewer's Responses to Questions

**Comments to the Authors:**

Reviewer #1: I thank the authors for their attention to my concerns, and those of the other reviewer, which were also very interesting. Although my original concrerns are appeased, a couple of new questions are now troubling me.

The authors write, l. 307 'To capture potential differences between individuals with MDD and healthy controls we ran the model separately for the two groups, using the same prior distributions.' If this is a hierarchical fit, running the model 'separately for the two groups' must mean that the model infers the group distributions for each group separately, based on the same priors on parameters describing these two distributions. Crucially, the consistency between each empirical prior and the parameters of each participant is also maximized, so that the parameters of each participant are informed indirectly by all the data of the group they belong to, but not the data of the other group. The authors then say (l. 321) that they 'test effects as the posterior distribution of difference between group posterior distributions and report the probability of one group having a higher estimated parameter value as the proportion of the distribution of difference above 0'. From this description, I am concerned whether shrinkage in the separate distributions of the two groups biases the second step towards false positives, as we and others some have shown in simulation studies. I would have thought that the correct way to avoid such an effect of shrinkage would be to include in the same model fitting separate distributions for the two groups, but also fit the difference between the means of these two distributions directly. Then, the credible interval of the difference directly gives what the authors seem to have estimated as a separate step. This is similar to a random effects fit where the data of the one group is allowed to inform the other group but only to the degree that the two groups are the same.

Secondly, I am concerned about the longitudinal analysis, and especially the claim that the follow-up data were 'completely out of sample'. I think the expression 'out of sample' is used in a rather different way, that is, that the test data are independent to with the training data (conditioned on group etc. of course). Here, the data are highly correlated by virtue of these being the same individuals. I'm not sure if the authors have done this already, but I think that two forms of out-of-sample prediction have to be distinguished here. The first is test-retest reliability, which is what the authors are doing. The same classifier that classifiers a participant at baseline also classifies them the same way at follow up. The second is true out-of-sample classification, where each follow-up datum is tested with a classifier *excluding that individual at all points where the classifier is trained*.

Thirdly, I am puzzled about the results that did not replicate at follow-up, or conversely, the new results that emerged then. Of course the main problem may be the small size of the sample, but I wonder if the authors could also discuss, in the discussion, the issues of state vs. trait and the role of the dimensional character of psychiatric disorder, and of course what happened in the follow-up period of six months. First, I observe that in this period three of the 10 participants who were originally classified as MDD no longer fulfilled such criteria, but they were included in analyses due to the small sample size. At first reading, this means that the classifier picked up some type of trait-like vulnerability or partial remission, as well as criteria-fulfilling MDD, or the results would be stronger without the remitters. Second, although the numbers are so small, one would expect more than 30% remission rate at 6 months - although of course these things vary enormously - but this 30% reminds the reader of the issue of self-selection. The retention rate for MDD is less than 1/2, which is is well below conventional satisfactory follow-up rates. Third, the remissions and composition of the follow-up sample suggest rather strongly that all results should be analysed dimensionally rather than categorically, using a measure of mood suitable for both clinical and health populations (if the authors have one). It would, for example, be very interesting if the classifier predicts status corrected for mood or indeed vice versa. Next, did the patients receive no treatment in these six months? Could it be simply treatment, rather than change of clinical status or randomness, that explains baseline-follow-up results?

Finally, I note that the authors very nicely included SHAPS following the suggestion of the other reviewer, but found no significant results. The authors should report a power analysis which will hopefully reassure us that the null result is simply due to lack of power. Though weak, the result is compatible with a view that sees anhedonia as a symptom of depression without as much interest with respect to vulnerability or endophenotype than the field hoped a few years ago.

Reviewer #2: The manuscript is much improved and I thank the authors for their thoughtful responses to my comments.

**Have all data underlying the figures and results presented in the manuscript been provided?**

Reviewer #1: Yes

Reviewer #2: **No: **They state restrictions, but will provide data upon request.

PLOS authors have the option to publish the peer review history of their article (what does this mean?). If published, this will include your full peer review and any attached files.

Reviewer #1: **Yes: **Dr. Michael Moutoussis

Reviewer #2: No
---

## [Decision Letter · Decision Letter 2]

9 Apr 2021

Dear Mr. Pedersen,

We are pleased to inform you that your manuscript 'Computational phenotyping of brain-behavior dynamics underlying approach-avoidance conflict in major depressive disorder' has been provisionally accepted for publication in PLOS Computational Biology.

Best regards,

Jean Daunizeau

Associate Editor

PLOS Computational Biology

Daniele Marinazzo

Deputy Editor

PLOS Computational Biology

Reviewer's Responses to Questions

**Comments to the Authors:**

Reviewer #1: I think that the authors' responsed to my concerns were satisfactory. I admit I rather enjoyed going through this and I would be very happy for this work to be published without delay.

**Have the authors made all data and (if applicable) computational code underlying the findings in their manuscript fully available?**

Reviewer #1: Yes

PLOS authors have the option to publish the peer review history of their article (what does this mean?). If published, this will include your full peer review and any attached files.

Reviewer #1: **Yes: **Michael Moutoussis

---

## [Editor Report · Acceptance letter]

5 May 2021

PCOMPBIOL-D-20-00326R2 

Computational phenotyping of brain-behavior dynamics underlying approach-avoidance conflict in major depressive disorder

Dear Dr Pedersen,

I am pleased to inform you that your manuscript has been formally accepted for publication in PLOS Computational Biology. Your manuscript is now with our production department and you will be notified of the publication date in due course.

With kind regards,

Katalin Szabo
